# PHYSICS-CONSTRAINED CORRELATION-AWARE ATTENTION FOR COLLECTIVE CELL DYNAMICS

**Dmitriy Stanislavchuk-Abovskii**
ITMO University,
Saint Petersburg, Russia
`dmitriy.abovskiy@gmail.com`

**Andrei Zakharov**
AXXX, Moscow, Russia
ITMO University,
Saint Petersburg, Russia
`andzkrv@gmail.com`

**Ilya Makarov**
AXXX, Moscow, Russia
ITMO University,
Saint Petersburg, Russia
`iamakarov@hse.ru`

## ABSTRACT

Collective cell migration is shaped by short-range physical interactions between neighboring cells, but trajectory predictors often rely on heuristic attention that lacks physical grounding and interpretability. We introduce a physics-constrained, correlation-aware attention framework that embeds analytically extracted direct interaction priors into graph-based models of cell dynamics. Our approach estimates the direct correlation function from empirical structure factors using Ornstein–Zernike-based deconvolution in Fourier space and incorporates this short-range signal into attention logits as a physical prior. To encourage consistent use of this prior, we propose a variational alignment objective that regularizes learned attention distributions toward physically motivated interaction patterns via a KL divergence. This framework yields physically meaningful attention representations and provides a principled way for integrating statistical physics into representation learning for biological dynamics. We present preliminary qualitative results on collective cell migration data and outline directions for systematic evaluation and extension.

## 1 INTRODUCTION

Collective cell migration underlies fundamental biological processes such as development, wound healing and cancer invasion. Time-lapse microscopy now routinely produces dense spatio-temporal recordings of many interacting cells, creating opportunities to learn predictive models of short-term cell motion directly from data.

We adopt the hypothesis that short-term trajectory prediction from time-lapse sequences can be formulated as the dynamics of an interacting-particle system. Concretely, we approximate cells as non-deformable interacting spheres. At each frame we observe cell positions and optional attributes, and the learning target is the one-step displacement between consecutive frames. Cell motion exhibits strong short-range correlations with nearby neighbors Cheung & Horne-Badovinac (2025), driven by chemical and mechanical signaling, kinetic effects and effective interaction potentials Zakharov & Dasbiswas (2021); Méhes & Vicsek (2014); Styer (2008).

To leverage this physical structure, we employ a graph neural network with attention on a radius graph and inject a data-derived interaction prior to bias the attention mechanism toward physically relevant short-range connections. Specifically, we estimate direct pairwise correlations from the empirical structure factor in $k$-space, inverse-transform the result to obtain a short-range kernel, and project these values onto graph edges to form a principled neighbor distribution that regularizes attention logits. This yields attention weights that can be interpreted as estimates of local interaction kernels, improving interpretability and sample efficiency. Unlike heuristic attention, this prior encodes experimentally measurable interaction statistics, providing an explicit physical inductive bias for learning collective dynamics. Although we demonstrate the approach on collective cell migration, the same framework generalizes to other interacting-particle domains such as animal groups, active matter, and multi-agent robotics.

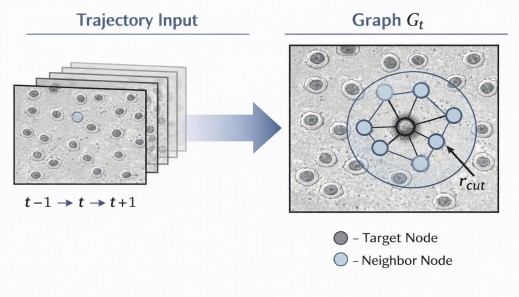

Figure 1: The figure schematically shows the cells detected at time $t$, represented as graph nodes, while edges connect neighboring cells within the cutoff radius $r_{\text{cut}}$. The shaded circle illustrates the area of short-range interaction used to determine the local neighborhood structure and pre-calculate the boundary based on physical data based on the direct correlation function $c(r)$.

## 2 RELATED WORK

Modern methods for trajectory prediction are formulated as a task of prediction by spatio-temporal networks, in which spatial dependencies are specified by an interaction graph, and temporal dynamics by recurrent models. For prioritizing interactions, self-attention is used, the addition of time information in spatiotemporal Transformer architectures allows one to effectively take into account local and global interactions between objects Emergent Mind (2025). Examples of typical approaches that combine graph operations and temporal modeling are STG-LSTM, as well as more modern spatio-temporal graph Transformer models (LITransformer, MTP-STG), demonstrating high quality in trajectory prediction tasks Ndunguru et al. (2025); Zhong et al. (2025); Zhang et al. (2025). However, such architectures remain computationally expensive and sensitive to the amount of data, which is usually compensated by training on large benchmarks, simulation, and aggressive data augmentation Zhong et al. (2025); Zhang et al. (2025).

In biomedical tasks, including cell trajectory prediction, only a limited number of video sequences is often available, which reduces the applicability of large-scale data-driven architectures and increases the risk of overfitting. Moreover, modern solutions do not have prioritization for physical informedness, which creates biases under limited data. This is consistent with the results of works where attention mechanisms are interpreted as an approximation of physical interactions during collective cell migration LaChance et al. (2022), as well as with approaches of physics-informed training that increase the stability and interpretability of dynamics models under small data volumes Jiang & Wan (2024).

## 3 DATA PREPARATION

After cell tracking, each detection has a track identifier TRACK_ID, frame index $t$ (FRAME) and coordinates $\boldsymbol{x}_i^t = (x_i^t, y_i^t)$, together with optional features $\boldsymbol{f}_i^t$ (radius, quality, etc.). From the detection table we create a training table by temporal differencing within tracks:

$$\boldsymbol{y}_i^t = \boldsymbol{x}_i^{t+1} - \boldsymbol{x}_i^t = (\Delta x_i^t, \Delta y_i^t), \tag{1}$$

where $\boldsymbol{y}_i^t$ is defined only if the track contains a point in frame $t+1$; this is taken into account by a validity mask $m_i^t \in \{0, 1\}$.

For each frame $t$ we construct a graph $G_t = (V_t, E_t)$, where $V_t$ is the set of detected cells in the frame, and edges are formed by the radius rule: $(i \rightarrow j) \in E_t$, if the Euclidean distance $r_{ij}^t = \|\boldsymbol{x}_i^t - \boldsymbol{x}_j^t\| \leq r_{\text{cut}}$ (see Fig. 1). The model input consists of node features $\boldsymbol{z}_i^t = [\boldsymbol{x}_i^t, \boldsymbol{f}_i^t]$ and edge features (for example, $r_{ij}^t$), and the output is the predicted displacement distribution for each node.

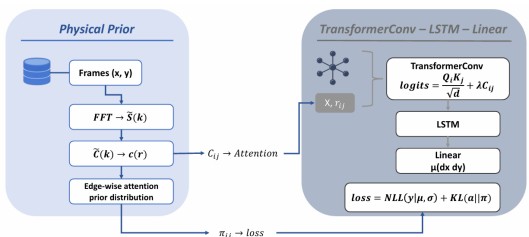

Figure 2: Architecture of the proposed network

## 4 COMPUTATION OF THE PHYSICAL PRIOR

To capture physically meaningful interactions, we use the direct correlation function $c(r)$ as a short-range interaction signal. Unlike the full correlation $h(r) = g(r) - 1$, it isolates direct pairwise contributions by removing indirect, multi-step effects mediated through third particles. In $k$-space, Fourier images $H(k)$ and $C(k)$ are considered, related through the structure factor $S(k)$:

$$S(k) = 1 + \rho H(k), \qquad C(k) = \frac{H(k)}{1 + \rho H(k)} = \frac{1 - \frac{1}{S(k)}}{\rho}, \tag{2}$$

where $\rho$ is the density. In practice, $S(k)$ is estimated from the observed coordinates using the Debye formula applied to frame subsamples. The resulting spectrum is stabilized via tail smoothing, after which $C(k)$ is computed using equation 2 and the real-space direct correlation $c(r)$ is recovered by the Hankel transform:

$$c(r) = \frac{1}{2\pi} \int_0^\infty k\, C(k)\, J_0(kr)\, dk, \tag{3}$$

where $J_0$ is the Bessel function of order zero. The computation $S(k) \to C(k) \to c(r)$ is performed outside the NN computational graph and cached, requiring no additional training cost. Hyperparameters of the prior estimation (effective field-of-view region, $k$ range, Hankel transform parameters, estimation of density $\rho$, and cutoff radius $r_{\mathrm{cut}}$) are selected automatically using the training frames.

Finally, $c(r)$ is projected onto graph edges: $c_{ij}^t = c(r_{ij}^t)$, $\tilde{C}_{ij}^t = \mathcal{N}(c_{ij}^t)$, where $\mathcal{N}$ is a robust normalization (centering, scaling, percentile clipping) to ensure a stable and comparable scale for the additive term in the attention logits.

## 5 INTEGRATION OF THE PRIOR INTO THE ARCHITECTURE

The model consists of two sequential graph attention blocks (modified `TransformerConv` Shi et al. (2020)) with nonlinearities, followed by an LSTM and a linear layer that predicts the displacement mean $\boldsymbol{\mu}_i^t \in \mathbb{R}^2$. The model architecture is shown in Fig. 2.

Graph attention is implemented using PyG Fey & Lenssen (2019), where attention normalization is performed via group softmax over incoming edges of the target node (indexed by $j$ for an edge $(i \to j)$). Therefore, the a priori neighbor distribution is consistently defined as:

$$\pi_{ij}^t = \mathrm{softmax}_{i \in \mathcal{N}(j)}\left(\gamma\, \tilde{C}_{ij}^t\right), \tag{4}$$

where $\gamma > 0$ controls the convexity of the prior. The prior $\pi$ is fixed and computed at the datamodule level, and then passed into each batch.

The network loss function is given by a Gaussian likelihood for displacements:

$$-\log p(\boldsymbol{y}_i^t \mid \boldsymbol{\mu}_i^t, \boldsymbol{\sigma}^2) = \frac{1}{2} \sum_{d \in \{x,y\}} \left[ \frac{(y_{i,d}^t - \mu_{i,d}^t)^2}{\sigma_{i,d}^2} + \log \sigma_{i,d}^2 \right] + \mathrm{const.} \tag{5}$$

In the simplest case, $\boldsymbol{\sigma}^2 = 1$, which is equivalent to a scaled MSE; however, this formalism allows heteroscedasticity when predicting $\log \boldsymbol{\sigma}^2$.

To align the attention distribution $\alpha_{ij}^{t,(h)}$ across heads $h$ with the physical prior equation 4, we added a regularization term:

$$\mathcal{L} = \underbrace{\mathbb{E}_{i,t}\Big[m_i^t \cdot \big(-\log p(\boldsymbol{y}_i^t \mid \boldsymbol{\mu}_i^t, \boldsymbol{\sigma}^2)\big)\Big]}_{\text{NLL over valid nodes}} + \eta_{\text{KL}} \underbrace{\mathbb{E}_{j,t}\left[\frac{1}{H}\sum_{h=1}^{H}\sum_{i\in\mathcal{N}(j)}\alpha_{ij}^{t,(h)}\log\frac{\alpha_{ij}^{t,(h)}}{\pi_{ij}^t}\right]}_{\text{KL}(\alpha\|\pi)\text{ over incoming edges}}, \qquad (6)$$

where $H$ is the number of heads. Averaging over target nodes $j$ prevents domination by nodes with a large number of neighbors; in practice, this is implemented by grouping terms by $j$ and computing the mean within each group. The coefficient $\eta_{\text{KL}}$ can be smoothly increased from zero to a target $\eta_{\text{KL}}^{\max}$ during training to avoid premature overconstraining of attention in the presence of noise and bias in the prior estimate. Setting $\eta_{\text{KL}}^{\max} = 0$, disables the KL regularization.

## 6 TEMPORAL DYNAMICS AND ROBUSTNESS TO SPARSE FRAMES

To model temporal dependence, we use an LSTM, where the hidden state is keyed by TRACK_ID and propagated across frames. This removes ambiguity in matching nodes between frames when the number of cells changes and allows the network to use the movement history of a particular cell preserving per-frame graph processing.

The initial frames of a video often contain only a few cells, leading to empty radius graphs. To prevent numerical failures in prior normalization and subsequent operations, the case $|E_t| = 0$ is handled by adding self-loop edges with zero values of $\tilde{C}_{ij}^t$ and zero distances. This maintains consistent tensor shapes and does not introduce an artificial signal.

## 7 RESULTS

We evaluate the proposed method on the BF-C2DL-HSC dataset from the Cell Tracking Challenge Cell Tracking Challenge (2026). Segmentation masks were obtained using Cellpose (v4) Stringer et al. (2021), which was fine-tuned and validated on manually annotated frames. Trajectories were constructed using TrackMate (the Fiji/ImageJ plugin) Ershov et al. (2022).

In the validation and test samples, the model demonstrates a consistent predictive performance, indicating stable generalization to unseen data (Fig. 3). For cell displacement components $(d_x, d_y)$, values of $R^2$ are in the range 0.65–0.75, with comparable error magnitudes $\text{RMSE}_{d_x} = 0.592$, $\text{RMSE}_{d_y} = 0.736$ on validation, and $\text{RMSE}_{d_x} = 0.638$, $\text{RMSE}_{d_y} = 0.688$ on test. Similar consistency is observed for the displacement magnitudes, with $\text{RMSE}_{|d|} \approx 0.94$ for both validation and test datasets, and $\text{MAE}_{|d|}$ is in the range of 0.69–0.73.

While we do not yet report comprehensive comparisons to baselines, these results suggest that the model captures meaningful variability in short-range cell motion and maintains performance across data splits, supporting the stability of the proposed physics-constrained attention framework.

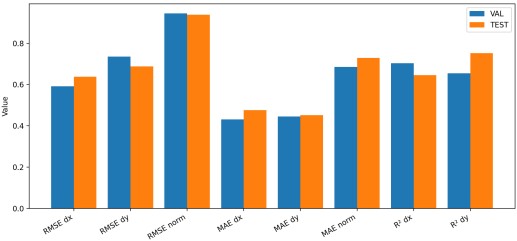

Figure 3: Comparison of prediction-quality metrics on validation vs. test splits

## 8 CONCLUSION

We propose a framework that integrates an out-of-network computed physical prior based on direct correlations $c(r)$ into graph attention mechanism via a consistent probabilistic formulation and regularization $\text{KL}(\alpha\|\pi)$. This approach enables the interpretability of attention weights, improves the signal-to-noise ratio in neighbor aggregation, and allows one to control balance between data-driven learning and physical inductive bias through the hyperparameters $\eta_{\text{KL}}$ and $\gamma$.

### MEANINGFULNESS STATEMENT

We contribute by extracting direct pairwise correlation spectra $C(k)$ from collective cell migration data via Ornstein–Zernike deconvolution, inverse-transforming to real space $c(r)$, and projecting it onto graph edges as a physics-derived prior. A variational KL alignment then regularizes learned attention $\alpha$ toward the prior $\pi$. The resulting attention-based embeddings explicitly parameterize short-range interaction kernels, yielding interpretable, inductive-bias-rich representations.

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
