# OpenReview forum: "Physics-Constrained Correlation-Aware Attention for Collective Cell Dynamics"
_ICLR.cc/2026/Workshop/LMRL — ICLR 2026 Workshop LMRL Poster_

### Official Review · Reviewer_FApx · 2026-02-15
**Attempt to predict cell movement via physical constraints**

**Rating:** 4
**Confidence:** 3

**Review:**

A trajectory prediction framework for live cell imaging data that uses the Ornstein-Zernike direct correlation function, derived from empirical structure factors, to bias graph attention weights toward physically probable cell interactions.

while the paper is clearly written, providing clear motivation and description of the suggestes approach it suffers from limited to no evaluation. Performance is presented on a single dataset, BF-C2DL-HSC, without any baseline comparison. Importantly, the single dataset has unique characteristics and does not represent "typical" cells in a dish--boundary and initial conditions restrict and dictate overall cell movement and interaction. Hence for example val/test splits are irrelevant as all cells are ~identical. Further, the method relies on accurate cell tracking data--which is not trivial to obtain.

To conclude, until the actual value and predictive power of the method is provided it is unclear what is the significance of this work.

---

### Official Review · Reviewer_Hch8 · 2026-02-16
**Physics-Guided Attention via Ornstein–Zernike Deconvolution for Collective Cell Dynamics**

**Rating:** 6
**Confidence:** 3

**Review:**

The paper proposes a physics-constrained, correlation-aware attention mechanism for predicting short-term collective cell dynamics on graphs. A direct correlation function c(r) is estimated from observed cell positions via an Ornstein–Zernike-based deconvolution of the structure factor S(k) in Fourier space, then projected onto graph edges to form a distance-based, short-range interaction prior π over neighbors; the learned attention is regularized toward this prior via a KL term, and c(r) is also described as an additive bias to attention logits. Preliminary experiments on a single Cell Tracking Challenge dataset report reasonable error metrics.

# Strengths

- The OZ-based pipeline to derive a physically motivated, short-range interaction prior for attention is an fresh and interesting idea for data-limited biological dynamics.

- The paper targets an important practical regime (small data) and, if validated, could yield more interpretable attention patterns and improved generalization by injecting a grounded inductive bias.


# Weaknesses and Questions

- Emphasis on short-range direct interactions may miss cases where long-range coordination or global fields dominate. The claim that multi-hop message passing and temporal modeling can recover these effects is plausible but not empirically tested.

- Please clarify the extent to which the OZ-derived c(r) should be interpreted as an effective statistical descriptor versus a causal interaction strength in active, nonequilibrium settings.

- How is k chosen in practice? How sensitive are c(r) and downstream performance to k?

- The prior estimation depends on multiple design choices (k, tail smoothing, density estimation, ROI selection). The paper lacks sensitivity analyses and robustness ablations for these components.

- The experimental section presents only preliminary results on a single dataset without systematic comparison to strong baselines  or ablations to isolate the contribution of the physical prior.

---

### Official Review · Reviewer_Q4EZ · 2026-02-26
**Interesting concept but insufficient results**

**Rating:** 4
**Confidence:** 2

**Review:**

Summary: The authors propose a method for integrating statistical mechanics-derived priors into graph attention to predict cell migration from microscopy data.

Strengths:

 - The general concept is well-framed and motivated.
- Physical priors could plausibly be useful in low-resource cell migration microscopy data.
- There is a clear use case and the authors identified a dataset.
- The authors anticipate some edge cases, e.g. empty radius graphs.

Weaknesses:

- The formulation is complex and brings together several niche areas, such that there is not much prior work to build off of. Other work on physical priors for attention is not discussed.
- Many specific additions or design choices do not feel well-motivated, especially without interpretable results.
- Even considering that preliminary results are acceptable, these results are difficult to interpret meaningfully. Without any baselines, we cannot draw any conclusions about the central claim of the paper, which assesses if physics-constrained attention improves performance.
- Simple ablations or baselines could measure if this approach provides any gains. As a suggestion, the authors could run simple experiments such as 1) omitting the additive term in attention logits and 2) setting the regularization weight eta=0, to check if those physics-informed additions provide any performance gain.

This is an interesting idea and paper, and although I do not believe it is sufficient in its current form, I would encourage authors to further investigate it. Some other suggestions:

- There seem to be strong modeling assumptions, which should be outlined more (including shape/property assumptions about cells and symmetry)
- More attention should be paid to the many hyperparams (r_cut, normalization of C_ij, gamma, eta), how they are chosen, and how they affect performance.
- The authors should clarify how the additive term C_ij is normalized and used, and how it is different from the prior, using equations rather than words.

---

### Meta-Review · Area_Chair_LXcU · 2026-02-28

**Recommendation:** Accept (Poster)
**Confidence:** 4

**Metareview:**

Accept.

---

### Decision · Program_Chairs · 2026-03-02

**Decision:**

Accept (Poster)

**Comment:**

Please see the meta-review.